# Socio-economic condition and lack of virological suppression among adults and adolescents receiving antiretroviral therapy in Ethiopia

**Martin Plymoth**[1]*, **Eduard J. Sanders**[2,3], **Elise M. Van Der Elst**[2], **Patrik Medstrand**[4], **Fregenet Tesfaye**[1], **Niclas Winqvist**[1], **Taye Balcha**[1], **Per Björkman**[1,5]

1 Clinical Infection Medicine, Department of Translational Medicine, Lund University, Malmö, Sweden,
2 Centre for Geographic Medicine Research, Kenya Medical Research Institute (KEMRI), Kilifi, Kenya,
3 Nuffield Department of Clinical Medicine, University of Oxford, Headington, United Kingdom, 4 Clinical Virology, Department of Translational Medicine, Lund University, Malmö, Sweden, 5 Department of Infectious Diseases, Skåne University Hospital, Malmö, Sweden

* martin.plymoth@gmail.com, martin.plymoth@norrbotten.se

**Data Availability Statement:** Data cannot be shared publicly because it contains sensitive information (including age, gender, relationship

## Abstract

### Introduction

The potential impact of socio-economic condition on virological suppression during antiretroviral treatment (ART) in sub-Saharan Africa is largely unknown. In this case-control study, we compared socio-economic factors among Ethiopian ART recipients with lack of virological suppression to those with undetectable viral load (VL).

### Methods

Cases (VL>1000 copies/ml) and controls (VL<150 copies/ml) aged $\geq$15years, with ART for >6 months and with available VL results within the last 3 months, were identified from registries at public ART clinics in Central Ethiopia. Questionnaire-based interviews on socio-economic characteristics, health condition and transmission risk behavior were conducted. Univariate variables associated with VL>1000 copies/ml (p<0.25) were added to a multivariable logistic regression model.

### Results

Among 307 participants (155 cases, 152 controls), 61.2% were female, and the median age was 38 years (IQR 32–46). Median HIV-RNA load among cases was 6,904 copies/ml (IQR 2,843–26,789). Compared to controls, cases were younger (median 36 vs. 39 years; p = 0.004), more likely to be male (46.5% vs. 30.9%; p = 0.005) and had lower pre-ART CD4 cell counts (170 vs. 220 cells/μl; p = 0.009). In multivariable analysis of urban residents (94.8%), VL>1000 copies/ml was associated with lower relative wealth (adjusted odds ratio [aOR] 2.98; 95% CI 1.49–5.94; p = 0.016), geographic work mobility (aOR 6.27, 95% CI 1.82–21.6; p = 0.016), younger age (aOR 0.94 [year], 95% CI 0.91–0.98; p = 0.011), longer duration of ART (aOR 1.19 [year], 95% CI 1.07–1.33; p = 0.020), and suboptimal (aOR

status, socio-economic condition, geographical location, and data on substance abuse and sexual behavior) which might lead to the identification of the study participant. Data are available through the Oromia Regional Health Bureau Ethics Committee (contact via Mr. Birhanu Kenate, BSc, MPH; email: simboyt@gmail.com; telephone: +251-911-043-813) for researchers who meet the criteria for access to confidential data.

**Funding:** The authors declare no conflicts of interest. This work was supported by the Swedish State under the agreement between the Swedish government and the county councils, the ALF-agreement [reference no 40103]; and a private donation to Lund University. PB was the recipient of both these grants. The funders had no role in study design, data collection and analysis, decision to publish, or preparation of the manuscript.

**Competing interests:** The authors have declared that no competing interests exist.

3.83, 95% CI 1.33–10.2; p = 0.048) or poor self-perceived wellbeing (aOR 9.75, 95% CI 2.85–33.4; p = 0.012), after correction for multiple comparisons. High-risk sexual behavior and substance use was not associated with lack of virological suppression.

## Conclusion

Geographic work mobility and lower relative wealth were associated with lack of virological suppression among Ethiopian ART recipients in this predominantly urban population. These characteristics indicate increased risk of treatment failure and the need for targeted interventions for persons with these risk factors.

## Introduction

By the end of 2017, 15.3 of an estimated 25.7 million people living with HIV (PLHIV) in sub-Saharan Africa were receiving antiretroviral therapy (ART) [1]. While ART results in suppressed HIV replication in most cases, many patients fail to achieve persistent virological suppression [2]. Incomplete virological suppression is commonly due to irregular drug intake and/or antiretroviral drug resistance [3, 4]. This compromises treatment outcomes for the individual, and also entails a risk of onward transmission, including dissemination of drug-resistant strains [5–7].

Lack of virological suppression during ART has been associated with advanced immuno-suppression, high baseline viral load, longer ART duration, younger age and male gender [8–13]. Failure to achieve virological suppression has also been linked to various socio-demographic factors, including unstable housing, low educational status, unemployment and lack of financial or social support [14–17]. These factors have been associated with disease outcomes in a range of both communicable and non-communicable conditions, and are considered as important determinants of health [18]. However, with regard to virologic response during ART, such relationships have mainly been investigated in high-income countries, and data is limited from high-burden low-income settings. For the long-term sustainability of ART programs, it is important to understand conditions contributing to lack of virological suppression, so that targeted interventions can be developed and implemented [7].

In many sub-Saharan African countries, the HIV epidemic is concentrated to certain geographic areas, as well as to key population groups. This is also the case for Ethiopia; whereas the overall national HIV prevalence is estimated at 0.9%, this figure is much higher in urban areas along the main highway transport routes [19–21]. Correspondingly, the estimated HIV prevalence among Ethiopian female commercial sex workers and long-distance truck drivers is high at 23% and 4.6%, respectively [22–24].

Similar to most other countries in sub-Saharan Africa, Ethiopia has implemented the World Health Organization (WHO) 'treat-all' policy [25]. Although treatment coverage has increased since ART delivery through the public health sector was introduced in 2005, less than 75% of PLHIV received ART in 2017 [19, 26, 27]. Furthermore, access to viral load (VL) testing for ART monitoring remains below 70% [7]. Cohort studies of patients starting ART in Ethiopia have shown that 20–30% of persons remaining in care after 6–12 months had one or several VL measurements >1000 copies/ml, implying high risk of treatment failure [12, 28, 29].

In this study, we aimed to explore the impact of socio-economic and demographic factors on virological suppression during ART. Since such factors might be associated with risk of

HIV transmission, we also investigated whether lack of virological suppression during ART is associated with high-risk sexual behavior.

## Methods

### Study setting

This study was performed at five public clinics providing ART, located in the city of Adama and surrounding towns in Central Ethiopia. The uptake area is located along the Addis Ababa-Djibouti highway, the main transport corridor in Ethiopia [20, 22, 23].

### Study design and procedure

Participants were identified from clinic registries, using a case-control design. These registries are separate for each study clinic, and contain data on time of ART initiation, ART regimens, CD4 cell counts, VL results, age and sex. All persons aged ≥15 years, with ART since ≥6 months and with a recorded VL result during the recent 3 months were eligible for inclusion. Cases were defined as patients with ≥1 measurements of HIV RNA >1000 copies/ml during the recent 3 months, whereas controls were required to have a recent VL measurement <150 copies/ml, and no recorded previous VL result exceeding this level.

For each person with a recorded VL result >1000 copies/ml found, at least one person with VL<150 copies/ml was selected randomly (without matching for age, sex or ART history) from the same registry as a potential control with the aid of a random number generator. Subsequently, previous recorded VL results were searched for these individuals. Since this procedure would lead to exclusion of some potential controls due to previous records of VL>1000 copies/ml, an excess of potential controls was obtained in the first step. For inclusion, access to individual medical records for retrieval of additional data was also required. After identification of potential study participants, these persons were contacted by phone and/or home visit and informed about the study, with an offer to return to the study clinic. At this visit, oral and written information was provided. Participants were included if written informed consent was obtained at this visit.

A minimum sample size (n = 294) for completed interviews was estimated using the Fleiss formula with continuity-correction based on an assumed prevalence of unstable employment and relative poverty of 50% among controls [30]; a two-sided confidence interval (CI) of 95%; a study power of 0.8; and a minimal detectable odds ratio (OR) of 2.0 [31].

Face-to-face interviews were conducted by MP and a trained facilitator with knowledge of the two main local languages (Amharic and Afan Oromo). Based on participant preference, the interviews were held in either of these languages in separate rooms at the respective study clinics between October and December 2018. The interviews followed a structured questionnaire translated from English into the two local languages and lasted approximately 30 minutes.

The questionnaire covered demographic and socio-economic characteristics (education level, employment, household income and wealth) and aspects of mental, physical and sexual health, including consumption of alcohol and khat (an amphetamine-like stimulant plant commonly used in Ethiopia). The following standardized survey tools were incorporated: the 15-question EquityTool Ethiopia (adapted and simplified from the 2016 Ethiopian Demographic and Health Survey [DHS]) in order to estimate relative wealth indices based on availability of items and services in households [32, 33]; the 2- and 9-question Patient Health Questionnaire-2 (PHQ)-2 and PHQ-9 translated into the local languages as screening for depressive disorder [34]; and the Fast Alcohol Screening Test (FAST) and Alcohol Use

Disorders Identification Test (AUDIT) as screening for harmful alcohol use [35]. For PHQ-2 score ≥3, the PHQ-9 questionnaire was applied, and for FAST scores ≥3, AUDIT was applied.

## Statistical analysis

Data were directly entered into Microsoft Excel. Data cleaning and analysis was performed using data analysis software SPSS 25.0 [36]. Unmatched univariate analysis was performed on continuous (Mann-Whitney U test) and categorical variables (logistic regression). For univariate analysis, a p-value <0.05 was considered as significant. Variables with p<0.25 were added to a multivariable logistic regression model. Variables showing multicollinearity (variance inflation factor [VIF]>5 in linear logistic regression including independent dummy categorical and continuous variables) or for which the proportion of missing data was >10%, as well as those that only concerned a minor subgroup of participants were excluded; this corresponded to the following variables: work type, recent CD4 count, AUDIT score and sexual behavior variables within the last month. The model was adapted through a stepwise backwards elimination protocol [37]. Variables with p<0.10 were kept in the model, as well as variables for which elimination led to >15% confounding effect on remaining variables. All eliminated variables were then re-entered, and those with p<0.15 were kept in the model. Holm-Šídák correction was used in the final model to determine statistical significant risk factors [38]. A separate analysis aggregated by sex was performed, comparing factors associated with VL>1000 copies/ml in univariate analysis.

## Ethical considerations

The study was approved by the Oromia Regional Health Bureau Ethical Review Committee. The research was conducted in accordance with the 2013 Declaration of Helsinki. All study participants received written and oral information about the study in either of the two local languages and provided written informed consent. For illiterate study participants, oral information was provided, and consent was confirmed through an impartial witness.

## Results

### Study participants

During the three-month period preceding initiation of the study, 5,025 VL results were recorded in the clinic registries from persons ≥15 years of age. Among these, 360 (7.2%) had HIV RNA>1000 copies/ml, whereas 4,537 (90.3%) had HIV RNA<150 copies/ml.

We identified 471 individuals (220 cases and 251 controls) based on 527 viral load results extracted from the VL registries. Medical records were retrieved for 431 (91.5%) of these (Fig 1). Subsequently, 378 subjects were reached by phone or home visit; of these, 307 (81.2%) accepted participation (155 cases and 152 controls).

### Participant characteristics

A majority of participants were female (61.2%), and the median age was 38 years (IQR 32–46). Compared to controls (Table 1), cases were younger (median 36 years vs. 39 years; p = 0.004) and more likely to be male (46.5% vs. 30.9%; p = 0.005). Cases had lower pre-ART and recent CD4 counts compared to controls (170 cells/μl vs. 220 cells/μl; p = 0.009, and 368 cells/μl vs. 518 cells/μl; p = 0.001, respectively). Furthermore, a higher proportion of cases received second-line ART (11.0% vs. 2.0%; p = 0.007).

Apart from higher pre-ART CD4 counts in the subset with VL>1000 copies/ml, no significant differences in characteristics were observed among persons who could not be considered

for inclusion due to lack of medical records (n = 40) or unsuccessful contact attempts (n = 53), as compared to included participants (S1 Table).

The median HIV RNA level among cases was 6,904 copies/ml (IQR 2,843–26,789). Sixty-four cases (41.8%) had VL>10,000 copies/ml, and 19 (12.4%) had VL>100,000 copies/ml.

## Sociodemographic characteristics

Data on socio-demographic characteristics of participants is shown in Table 2. In terms of civil status, cases were more likely to be single or divorced (22.6% vs. 11.2% and 28.4% vs. 20.4%, respectively; p = 0.006). Ethnicity, religion or education level did not differ between groups, nor did number or type of languages spoken.

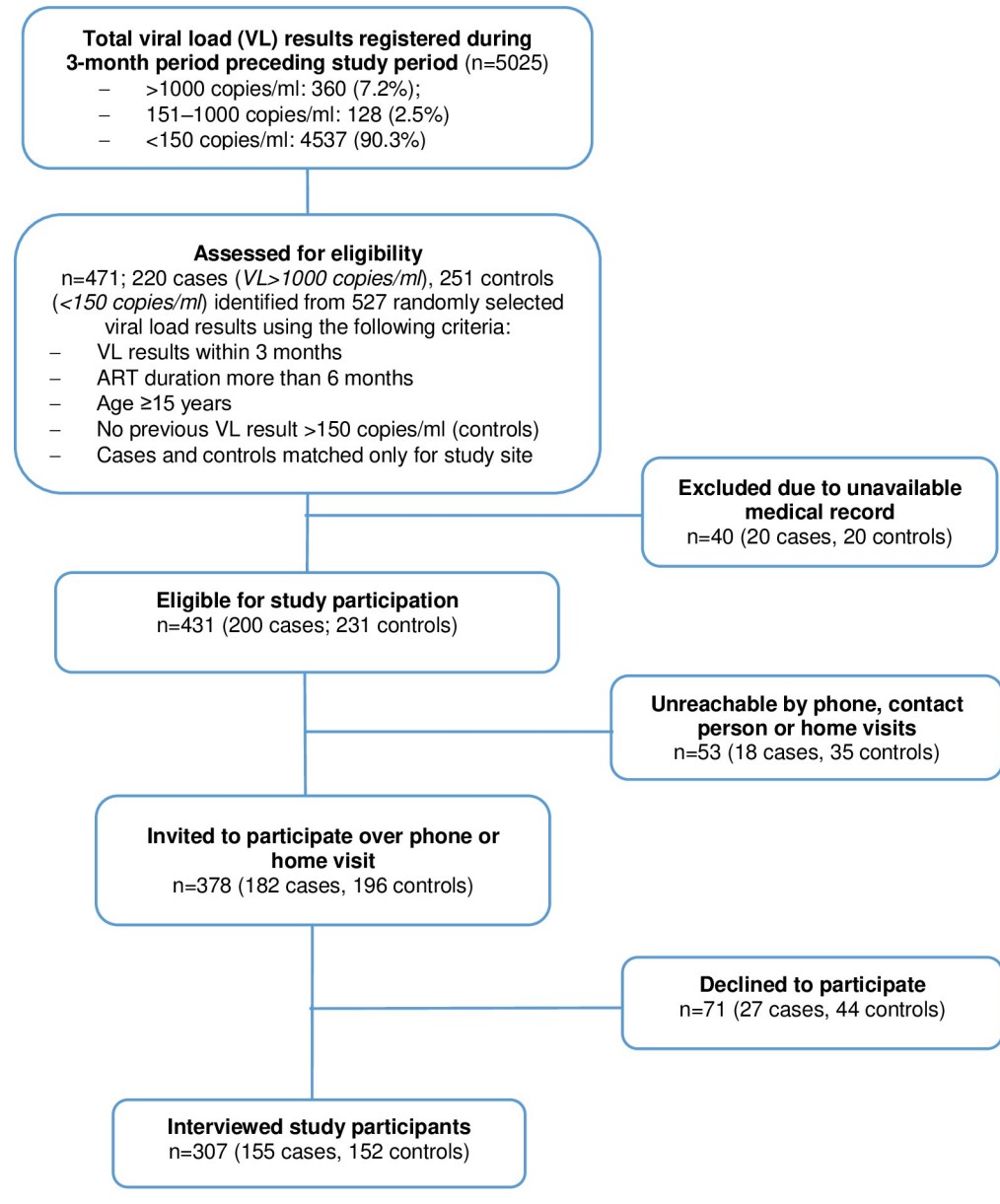

VL: Viral load; n: Number; ART: Antiretroviral therapy.

**Fig 1. Flowchart of study participant recruitment.**

**Table 1. Comparison of demographic and medical characteristics of study participants on ART with lack of virological suppression and with undetectable viral load.**

| | Unsuppressed VL (>1000 copies/ml; n = 155) | Undetectable VL (<150 copies/ml; n = 152) | Odds Ratio (95% CI) | p-value |
|---|---|---|---|---|
| **Age (years; median, IQR)** | 36 (30–44) | 39 (34–48) | | **0.004** |
| **Gender** | | | | |
| Male | 72 (46.5) | 47 (30.9) | **1.94 (1.22–3.10)** | **0.005** |
| Female | 83 (53.5) | 105 (69.1) | 1.00 | |
| Pre-ART CD4 count[†] (cells/ mm$^3$; median, IQR) | 170 (90–274) | 220 (124–357) | | **0.009** |
| Recent CD4 count, <1 year[‡] (cells/ mm$^3$; median, IQR) | 368 (275–454) | 518 (371–617) | | **0.001** |
| **Duration of ART (years; median, IQR)** | 8.7 (6.4–10.9) | 7.8 (4.6–10.2) | | **0.027** |
| ART regimen[§] | | | | **0.007** |
| First-line | 137 (89.0) | 149 (98.0) | 1.00 | |
| Second-line | 17 (11.0) | 3 (2.0)[¶] | **6.17 (1.77–21.3)** | |
| **Transfer-in to current ART clinic** | | | | 0.129 |
| Yes | 28 (18.2) | 18 (11.8) | 1.64 (0.87–3.11) | |
| No | 127 (81.8) | 147 (88.2) | 1.00 | |
| **ART clinic location** | | | | 0.169 |
| Same district as residence | 128 (82.6) | 131 (87.6) | 1.00 | |
| Different district | 27 (17.4) | 21 (12.4) | 1.57 (0.83–2.99) | |
| **Viral load (copies/ml; median, IQR)** | 6904 (2843–26789) | <150 | | |
| 1000–10,000 | 89 (58.2) | | | |
| 10,000–100,000 | 45 (29.4) | | | |
| >100,000 | 19 (12.4) | | | |

Binominal logistic regression for categorical variables. Mann-Whitney U test for continuous variables. Data reported in absolute number and percentage (in brackets) and unadjusted odds ratios, unless otherwise specified. VL: viral load; CD4: CD4 cell count; IQR: Interquartile range; ART: Antiretroviral therapy.

† Missing values of pre-ART CD4: 20 (cases); 10 (controls)

‡ Missing values of recent CD4 (within 1 year): 120 (cases); 111 (controls)

§ First-line regimens (based on non-nucleoside reverse transcriptase inhibitors): Nevirapine-based: 40.9% (cases), 37.5% (controls); Efavirenz-based: 48.0% (cases), 60.5% (controls); Second-line regimens (based on protease inhibitors): Atazanavir/ritonavir-based: 9.0% (cases), 1.3% (controls); Lopinavir/ritonavir-based: 1.3% (cases), 0.7% (controls)

¶ No previous detectable VL results found

With regard to employment status, higher proportions of cases than controls were daily laborers or students (25.8% vs. 13.8% and 11.0% vs. 2.6%, respectively; p = 0.001). Construction work (15.5% vs. 5.9%; p = 0.013), working in multiple locations outside residential district (11.0% vs. 3.9%; p = 0.021) and travelling home from workplace less frequently than daily (11.6% vs. 4.6%; p = 0.030) was more commonly reported among cases than controls.

Self-reported household income was lower among cases, with 64.5% reporting <35 USD (United States Dollar) per month (64.5% vs. 46.7%; p = 0.002). Urban wealth quintiles (i.e. wealth distribution divided into fifths from poorest to richest) were calculated for non-rural residents (94.8% of participants). The lower three quintiles were overrepresented among cases as compared to controls (70.5% vs. 49.0%; p<0.001); while the greatest difference was observed among the lowest quintile (11.0% vs 4.8%; p = 0.003).

## Physical, mental and sexual health characteristics

The majority of participants rated their state of wellbeing as excellent, with no period of prolonged (≥4 days) illness during the preceding month (Table 3). However, worse perceived

**Table 2. Comparison of sociodemographic characteristics of study participants on ART with lack of virological suppression and with undetectable viral load.**

| | Unsuppressed VL (>1000 copies/ml; n = 155) | Undetectable VL (<150 copies/ml; n = 152) | Odds ratio (95% CI) | p-value |
|---|---|---|---|---|
| **Civil status** | | | | **0.006** |
| Married | 52 (33.5) | 70 (46.1) | 1.00 | |
| Single | 35 (22.6) | 17 (11.2) | **2.77 (1.40–5.48)** | |
| Divorced | 44 (28.4) | 31 (20.4) | **1.91 (1.07–3.42)** | |
| Widowed | 24 (15.5) | 34 (22.4) | 0.95 (0.50–1.79) | |
| **Ethnicity** | | | | 0.492 |
| Oromo | 78 (50.3) | 67 (44.1) | 1.00 | |
| Amhara | 48 (31.0) | 56 (36.8) | 0.74 (0.44–1.22) | |
| Other | 29 (18.7) | 29 (19.1) | 0.86 (0.47–1.58) | |
| **Number of languages spoken (median, IQR)** | 2 (1–2) | 2 (1–2) | | 0.704 |
| **Religion** | | | | 0.734 |
| Orthodox | 113 (72.9) | 106 (69.7) | 1.00 | |
| Protestant | 28 (18.1) | 33 (21.7) | 0.80 (0.45–1.41) | |
| Muslim | 13 (8.4) | 13 (8.6) | 0.94 (0.42–2.12) | |
| Other | 1 (0.6) | 0 | N/A | |
| **Education** | | | | 0.959 |
| No education | 31 (20.0) | 34 (22.4) | 1.00 | |
| Primary (1–8) | 71 (45.8) | 66 (43.3) | 1.18 (0.65–2.13) | |
| Secondary (9–12) | 39 (25.2) | 38 (25.0) | 1.13 (0.58–2.18) | |
| Tertiary[†] | 14 (9.0) | 14 (9.2) | 1.10 (0.45–2.66) | |
| **Employment status** | | | | **0.001** |
| Formal employment | 36 (23.3) | 50 (32.9) | 1.00 | |
| Self-employed | 35 (22.6) | 31 (20.4) | 1.57 (0.82–2.99) | |
| Daily labor | 40 (25.8) | 21 (13.8) | **2.65 (1.34–5.22)** | |
| Housewife | 14 (9.0) | 28 (18.4) | 0.69 (0.32–1.50) | |
| Student | 17 (11.0) | 4 (2.6) | **5.90 (1.83–19.0)** | |
| Unemployed[‡] | 13 (8.4) | 18 (11.8) | 1.00 (0.44–2.31) | |
| **Job security** | | | | 0.057 |
| Seldom or never concerned | 123 (79.4) | 133 (87.5) | 1.00 | |
| Regularly concerned | 32 (20.6) | 19 (12.5) | 1.82 (0.98–3.38) | |
| **Work type[§]** | | | | **0.013** |
| No work | 31 (20.0) | 31 (20.4) | 1.00 | |
| Agriculture | 10 (6.5) | 8 (5.3) | 1.23 (0.44–3.59) | |
| Transportation | 13 (8.4) | 4 (2.6) | 3.25 (0.95–11.1) | |
| Retail/petty trade | 39 (25.2) | 34 (22.4) | 1.15 (0.58–2.26) | |
| Public office/education | 3 (1.9) | 9 (5.9) | 0.33 (0.08–1.35) | |
| Security/military | 9 (5.8) | 8 (5.3) | 1.13 (0.38–3.30) | |
| Construction | 24 (15.5) | 9 (5.9) | **2.67 (1.07–6.65)** | |
| Housework/cleaning | 14 (9.0) | 24 (15.8) | 0.58 (0.26–1.33) | |
| Industry/skilled labor | 6 (3.9) | 11 (7.2) | 0.55 (0.18–1.66) | |
| Other professions[¶] | 6 (3.9) | 14 (9.9) | 0.43 (0.15–1.26) | |
| **Job Location** | | | | **0.021** |
| Within residential district | 123 (79.4) | 138 (90.8) | 1.00 | |
| Single location outside residential district | 15 (9.7) | 8 (11.4) | 2.01 (0.86–5.13) | |
| Multiple locations outside residential district | 17 (11.0) | 6 (3.9) | **3.18 (1.22–8.32)** | |

*(Continued)*

**Table 2.** (Continued)

| | Unsuppressed VL (>1000 copies/ml; n = 155) | Undetectable VL (<150 copies/ml; n = 152) | Odds ratio (95% CI) | p-value |
|---|---|---|---|---|
| *Household* | | | | |
| **Job commute** | | | | **0.030** |
| Home every day | 137 (88.4) | 145 (95.4) | 1.00 | |
| Home less often | 18 (11.6) | 7 (4.6) | **2.72 (1.10–6.72)** | |
| **Monthly household income** | | | | **0.002** |
| <35 USD | 100 (64.5) | 71 (46.7) | **2.07 (1.31–3.28)** | |
| ≥35 USD | 55 (35.5) | 81 (53.3) | 1.00 | |
| **Residence ownership** | | | | 0.154 |
| Owned | 43 (27.7) | 46 (30.3) | 1.00 | |
| Rented | 81 (52.3) | 88 (57.9) | 0.99 (0.59–1.65) | |
| Owned by family/others | 31 (20.0) | 18 (11.8) | 1.84 (0.90–3.76) | |
| **Number of rooms (median, IQR)** | 2 (1–3) | 2 (1–2) | | 0.373 |
| **Household members (median, IQR)** | 3 (2–4) | 3 (2–4) | | 0.785 |
| **Number of children (median, IQR)** | 2 (0–3) | 2 (1–3) | | 0.188 |
| **Residence in rural area** | | | | 0.637 |
| Yes | 9 (5.8) | 7 (4.6) | 1.28 (0.46–3.52) | |
| No | 146 (94.2) | 145 (95.4) | 1.00 | |
| Urban wealth quintile[††] | | | | **0.003** |
| 1st (poorest) | 16 (11.0) | 7 (4.8) | **3.62 (1.30–10.1)** | |
| 2nd | 33 (22.6) | 25 (17.2) | **2.09 (1.01–4.33)** | |
| 3rd | 54 (37.0) | 38 (26.2) | **2.25 (1.17–4.35)** | |
| 4th | 19 (13.0) | 37 (25.5) | 0.81 (0.38–1.73) | |
| 5th (richest) | 24 (16.4) | 38 (26.2) | 1.00 | |
| **Lower three urban wealth quintiles** | 103 (70.5) | 70 (48.3) | **2.57 (1.58–4.16)** | **<0.001** |

Binominal logistic regression for categorical variables. Mann-Whitney U test for continuous variables. Data reported in absolute number and percentage (in brackets) and unadjusted odds ratios, unless otherwise specified. VL: viral load; IQR: Interquartile range; USD: United States Dollar.

† Including Technical and Vocational Education and Training (TVET) and university degrees

‡ Including unpaid volunteers work and pension/retirement

§ Including previous work for people currently unemployed

ⱹ Commercial sex work (n = 2), finance (n = 3), healthcare (n = 3), server (n = 5), (assistant) chef (n = 4), cosmetics and hair-care (n = 2)

†† Sub analysis on urban study population: cases (n = 146) and controls (n = 145)

wellbeing (p<0.001) and self-reported illness lasting >3 days (14.8% vs. 4.6%; p = 0.004) was more common among cases than controls. At least one episode of anhedonia or depressed mood within a 2-week period was reported among 18 cases and 13 controls (11.6% vs. 8.7%; p = 0.375).

Twenty-nine participants reported that they had never had a sexual contact (21 cases, 7 controls). The median age of these persons was 18 years (IQR 17.3–19.0), and all of them had acquired HIV vertically. For sexually active persons, no statistically significant association between VL category and number of lifetime sexual partners, age at sexual debut and history of sexually transmitted infections was observed. The proportion of persons reporting sexual activity during the recent month was lower among cases than controls (17.4% vs. 25.7%; p = 0.028). Among these, there was no significant difference in terms of rates of discordant partners, condom use or disclosure to partner.

Although alcohol consumption was reported in similar proportions of cases and controls (35.5% vs. 30.9%; p = 0.721), hazardous alcohol use was more common among cases (FAST

**Table 3. Comparison of physical, mental and sexual health, and substance use of study participants on ART with lack of virological suppression and with undetectable viral load.**

| | Unsuppressed VL (>1000 copies/ml; n = 155) | Undetectable VL (<150 copies/ml; n = 152) | Odds ratio (95% CI) | p-value |
|---|---|---|---|---|
| **Self-rated wellbeing** | | | | **<0.001** |
| Excellent | 102 (65.8) | 131 (86.2) | 1.00 | |
| Suboptimal | 31 (20.0) | 12 (7.9) | **3.32 (1.62–6.78)** | |
| Poor | 20 (12.9) | 7 (4.7) | **3.14 (1.62–6.78)** | |
| **Self-reported illness last month** | | | | **0.004** |
| 0–3 days | 132 (85.2) | 145 (95.4) | 1.00 | |
| ≥4 days | 23 (14.8) | 7 (4.6) | **3.61 (1.50–8.69)** | |
| **Depressive symptoms (PHQ-2 score)** | | | | 0.375 |
| 0 | 137 (88.4) | 136 (91.3) | 1.00 | |
| ≥1 | 18 (11.6) | 13 (8.7) | 1.41 (0.66–2.98) | |
| **Major depressive disorder (PHQ-9 score ≥8; n = 5 [PHQ-2 score ≥3])** | 2 (1.3) | 1 (0.6) | | |
| *Sexual health* | | | | |
| **Number of lifetime sexual partners** | | | | **0.020** |
| 1–5 | 91 (58.7) | 111 (73.0) | 1.00 | |
| 6–10 | 12 (7.7) | 7 (4.6) | 2.09 (0.79–5.53) | |
| >10 | 31 (20.0) | 27 (17.8) | 1.40 (0.78–2.52) | |
| Never sexually active | 21 (13.5) | 7 (4.6) | **3.66 (1.49–8.99)** | |
| **Age at sexual debut (median, IQR)** | 18 (15.5–20) | 18 (15–20) | | 0.724 |
| History of STI[†] | | | | 0.444 |
| Yes | 34 (21.9) | 39 (27.4) | 0.81 (0.48–1.38) | |
| No | 121 (78.1) | 113 (72.6) | 1.00 | |
| **Sexually active, last month** | | | | **0.028** |
| Yes | 27 (17.4) | 47 (25.7) | **0.54 (0.31–0.94)** | |
| No | 128 (82.6) | 105 (74.3) | 1.00 | |
| **>1 sexual partner, last month** | *n = 27* | *n = 47* | N/A | |
| Yes | 2 (7.4) | 0 | | |
| No | 25 (92.6) | 47 (100) | | |
| **Discordant partner, last month** | *n = 27* | *n = 47* | | 0.207 |
| Yes | 8 (29.7) | 14 (29.8) | 1.00 | |
| No | 13 (48.1) | 26 (55.3) | 1.14 (0.38–3.41) | |
| Unknown | 6 (22.2) | 7 (14.9) | 0.57 (0.14–2.38) | |
| **Condom use, last month** | *n = 27* | *n = 47* | | 0.283 |
| Always | 14 (51.9) | 28 (59.6) | 1.00 | |
| Less often | 13 (48.1) | 19 (40.4) | 1.71 (0.64–4.53) | |
| **HIV disclosure to partner, last month** | *n = 27* | *n = 47* | | 0.275 |
| Yes | 24 (88.9) | 45 (95.7) | 1.00 | |
| No | 3 (11.1) | 2 (4.3) | 2.81 (0.44–18.0) | |
| *Substance use* | | | | |
| **Alcohol consumption** | | | | 0.721 |
| Less than once a year | 100 (64.5) | 105 (69.1) | 1.00 | |
| Once a month or less | 34 (21.9) | 32 (20.1) | 1.12 (0.64–1.94) | |
| 2–4 times a month | 11 (7.1) | 9 (6.0) | 1.28 (0.51–3.23) | |
| 2 or more times a week | 10 (6.5) | 6 (4.0) | 1.75 (0.61–4.99) | |
| **Hazardous drinking (FAST Score)** | | | | **0.006** |
| Yes (≥3) | 22 (14.2) | 7 (4.6) | **3.43 (1.42–8.28)** | |

*(Continued)*

**Table 3.** (Continued)

| | Unsuppressed VL (>1000 copies/ml; n = 155) | Undetectable VL (<150 copies/ml; n = 152) | Odds ratio (95% CI) | p-value |
|---|---|---|---|---|
| No (<3) | 133 (85.8) | 145 (93.4) | 1.00 | |
| Harmful alcohol use (n = 29 [FAST score ≥3]; AUDIT score ≥8) | 9 (5.8) | 3 (2.0) | 3.06 (0.81–11.5) | 0.098 |
| Regular *Khat* use‡ | | | | **0.030** |
| Yes | 18 (11.6) | 7 (4.6) | **2.72 (1.10–6.72)** | |
| No | 137 (88.4) | 145 (95.4) | 1.00 | |

Binominal logistic regression for categorical variables. Mann-Whitney U test for continuous variables. Data reported in absolute number and percentage (in brackets) and unadjusted odds ratios, unless otherwise specified. VL: viral load; IQR: Interquartile range; PHQ: Patient Health Questionnaire; STI: Sexually Transmitted Infection; FAST: Fast Alcohol Screening Test; AUDIT: Alcohol Use Disorder Identification Test.

† History of symptoms of gonorrhea-like discharge, chancre-like lesion and/or genital itchiness

‡ An amphetamine-like plant substance ingested through chewing. Regular defined as on consumption at repeated occasions within the last year

score ≥3; 14.2% vs. 4.6%; p = 0.006). Further assessment using the AUDIT score did not show significant differences with regard to harmful alcohol use and VL category (AUDIT score ≥8; 5.8% vs. 2.0% in cases and controls, respectively; p = 0.098). Furthermore, khat use was more commonly reported among cases (11.6% vs. 4.6%; p = 0.030).

## Factors associated with VL>1000 copies/ml in multivariable analysis

Four variables were found to be associated with VL>1000 copies/ml (p<0.05) in multivariable analysis (Table 4) but were not significant after Holm-Šídák correction (daily labor, monthly household income <35 USD, pre-ART CD4 cell count and hazardous alcohol use). After correction for multiple comparisons, younger age (aOR 0.94 [per year], 95% CI 0.91–0.98; p = 0.011); having longer ART duration (aOR [per year] 1.19, 95% CI 1.07–1.33; p = 0.020); working at multiple locations outside residential district (aOR 5.66, 95% CI 1.67–19.2; p = 0.030); reporting suboptimal (aOR 3.83, 95% CI 1.33–10.2; p = 0.048) or poor self-perceived wellbeing (aOR 9.75, 95% CI 2.85–33.4; p = 0.012); and belonging to a lower urban wealth quintile (aOR 2.98, 95% CI 1.49–5.94; p = 0.016) remained independently associated with high VL.

## Gender-specific factors associated with VL>1000 copies/ml

Although the design of our study was not powered for sub-group analysis, univariate analysis of the data disaggregated by gender showed several associations with virological outcome during ART (S2–S4 Tables).

Among both men and women, VL>1000 copies/ml was associated with lower relative wealth (71.2% vs. 46.5%; p = 0.011 and 70.0% vs. 50.0%; p = 0.007, respectively), whereas geographic work mobility did not show such association in gender-specific analysis. Younger age was associated with unsuppressed viral load only among men (38.0 vs. 47.0 years; p<0.001), while poor reported well-being showed such an association only among women (21.7% vs. 8.6%; p = 0.004).

In addition, the following variables were found to be more common in cases than controls among men only: hazardous alcohol drinking (FAST score≥3; 23.6% vs. 8.5%; p = 0.042); single (33.3% vs. 19.1%; p = 0.022) or divorced (23.6% vs. 8.5%; p = 0.012) civil status; lack of ownership or rental of household (23.6% vs. 6.4%; p = 0.006); having fewer children (median 1 vs. 2; p = 0.005); and younger age at sexual debut (median 18 vs. 20 years; p = 0.040).

Furthermore, as compared to male controls, male cases were less likely to report multiple lifetime sex partners (n>10; 15.3% vs. 44.7%; p = 0.003), previous sexually transmitted

**Table 4. Factors associated with lack of viral suppression among urban study participants, Adama Region, Ethiopia.**

| | Crude Odds Ratio (95% CI) | Univariate p-value | Adjusted Odds Ratio (95% CI) | Multivariable p-value | Adjusted p-value§ |
|---|---|---|---|---|---|
| **Age (years)** | 0.97 (0.95–0.99) | 0.004 | 0.94 (0.91–0.98) | 0.001 | **0.011** |
| **Gender** | | 0.005 | | 0.058 | |
| Male | 1.94 (1.22–3.10) | | 2.15 (0.97–4.75) | | |
| Female | 1.00 | | 1.00 | | |
| Pre-ART CD4 count ($10^{-1}$ cells/mm$^3$)† | 0.98 (0.97–1.00) | 0.009 | 0.98 (0.96–1.00) | 0.033 | |
| **Duration of ART (years)** | 1.07 (1.00–1.14) | 0.007 | 1.19 (1.07–1.33) | 0.002 | **0.020** |
| ART regimen‡ | | 0.007 | | 0.139 | |
| First-line | 1.00 | | 1.00 | | |
| Second-line | 6.17 (1.77–21.3) | | 3.33 (0.68–16.4) | | |
| **Job location** | | 0.021 | | 0.009 | |
| Within residential district | 1.00 | | 1.00 | Ref | |
| Single location outside residential district | 2.01 (0.86–5.13) | | 2.50 (0.77–8.16) | 0.128 | |
| Multiple locations outside residential district | 3.18 (1.22–8.32) | | 6.27 (1.82–21.6) | 0.004 | **0.032** |
| **Employment status** | | 0.001 | | 0.375 | |
| Formal employment | 1.00 | | 1.00 | Ref | |
| Self-employed | 1.57 (0.82–2.99) | | 1.74 (0.70–4.33) | 0.236 | |
| Daily labor | 2.65 (1.34–5.22) | | 2.97 (1.08–8.16) | 0.035 | |
| Housewife | 0.69 (0.32–1.50) | | 1.24 (0.38–4.02) | 0.720 | |
| Student | 5.90 (1.83–19.0) | | 1.15 (0.20–6.72) | 0.873 | |
| Unemployed | 1.00 (0.44–2.31) | | 1.12 (0.37–3.88) | 0.763 | |
| **Monthly household income** | | 0.002 | | 0.024 | |
| ≥35 USD | 1.00 | | 1.00 | | |
| <35 USD | 2.07 (1.31–3.28) | | 2.33 (1.12–4.84) | | |
| **Urban wealth quintile** | | <0.001 | | 0.002 | **0.016** |
| 4th-5th (wealthiest) | 1.00 | | 1.00 | | |
| 1st-3rd (poorest) | 2.57 (1.58–4.16) | | 2.98 (1.49–5.94) | | |
| **Wellbeing** | | <0.001 | | <0.001 | |
| Excellent | 1.00 | | 1.00 | Ref | |
| Suboptimal | 3.32 (1.62–6.78) | | 3.83 (1.44–10.2) | 0.007 | **0.048** |
| Poor | 3.14 (1.62–6.78) | | 9.75 (2.85–33.4) | <0.001 | **0.012** |
| **Hazardous alcohol use (FAST score)** | | 0.006 | | 0.045 | |
| No (<3) | 1.00 | | 1.00 | | |
| Yes (≥3) | 3.43 (1.42–8.28) | | 3.60 (1.03–12.6) | | |
| **Residence ownership** | | 0.154 | | 0.430 | |
| Owned | 1.00 | | 1.00 | | |
| Rented | 0.99 (0.59–1.65) | | 0.76 (0.35–1.67) | 0.494 | |
| Owned by family/others | 1.84 (0.90–3.76) | | 0.46(0.14–1.48) | 0.195 | |

Multivariable logistic regression model of variables associated with high viral load results, showing remaining variables with a p-value <0.15. Data reported in crude and adjusted odds ratios, univariate and multivariable p-values and significance after Holm-Šídák correction. Due to the inclusion of urban wealth quintiles in the model, only urban residents (n = 291) were considered. Due to missing values, 260 (89.7%) study participants were included in the model. In stepwise backwards elimination protocol the following variables were excluded from the model: Transfer-in to current ART clinic; civil status; job security; job commute; number of children; Self-reported illness (last month); ART clinic location; regular khat use. CD4: CD4 cell count; USD: United States Dollar; FAST: Fast Alcohol Screening Test.

† Missing values n = 30

‡ Missing values n = 1

§ Significant adjusted p-values after Holm-Šídák correction (marked in bold)

infection (22.2% vs. 57.4%; p<0.001), and recent sexual activity (15.5% vs. 39.1%; p = 0.008), respectively. Among women, report of multiple lifetime sexual partners (24.1% vs. 5.7%; p<0.001) and longer ART duration (9.31 vs. 7.76 years; p = 0.003) was more common among cases than controls.

## Discussion

In this case-control study conducted at Ethiopian ART clinics, lack of virological suppression during ART was independently associated with two socio-economic factors: geographic work mobility and lower relative wealth in urban residents.

In high-income countries, indicators of socio-economic disadvantage have been associated with inadequate virological suppression during ART. Detectable VL was more common among ART recipients in the United Kingdom with unstable housing, non-university education, unemployment and financial hardship [15]. Other social and structural factors, such as homelessness, injection drug use and lack of social support, have also been associated with viremia during ART [16, 17].

Although rates of virological suppression have been reported to be similar in cohorts receiving ART in high- and low-income countries [39], our findings suggest that socio-economic conditions influence virological treatment response in low-income countries. Irregular or interrupted therapy has been linked to multiple geographical, social and economic factors in resource-constrained settings [40]. Indirect costs related to ART, such as travel costs and absence from work, are examples of such barriers to adherence [41]. Similar associations between socio-economic vulnerability and worse treatment outcomes have been reported for tuberculosis (including multi-drug resistant tuberculosis) and malaria [42–45].

Data on associations between socio-economic factors and virological ART outcomes from sub-Saharan Africa is limited, and findings are discordant [46]. While no clear association between household income and virological outcomes during ART was reported in a study from South Africa, higher income was associated with virological treatment failure in a recent study from Uganda [47, 48]. In contrast, a large study conducted in Uganda and Kenya 2013–2014 showed an association between lower household income and worse virological outcomes [49].

In our study we used both a direct estimate of household income and an established tool for assessment of relative wealth (EquityTool) to characterize economic status [33]. Defining relevant cut-offs for household income requires consideration of both dynamic urban-rural as well as country-specific conditions. This holds especially true for Ethiopia which has the fastest growing economy in sub-Saharan Africa, but where around 24% of the population still live in poverty [50]. We used a monthly household income of 35 USD as a threshold in our study, corresponding to the population living on less than 1.25 USD per day [51].

To our knowledge, this is the first study that employs a simplified and comprehensive tool for assessment of relative wealth in relation to virological outcome in the ART "treat-all" era. In line with a previous Ethiopian study showing an association between low household income and treatment adherence [52], we found that persons with lower relative wealth were more likely to have lack of virological suppression during ART. We further showed relative wealth to be more strongly associated with virological outcomes than household income alone.

Besides low relative wealth, incomplete virological suppression was associated with work in multiple locations outside place of residence. Although the underlying mechanisms could not be elucidated with our study design, it is possible that the requirements of current ART programs in Ethiopia constitute an obstacle for adherence for persons with mobile work.

Migrant and mobile populations are recognized to be at increased risk of HIV acquisition [53–58]. Linkage and retention in care is also often inadequate in such populations [59]. In

agreement with this, studies from different parts of sub-Saharan Africa have shown that non-residents on ART are less likely to have virological suppression than residents [60, 61].

Although our study did not specifically target migrants, the finding of lower rates of virological suppression in mobile workers provides further support for the role of mobility for ART outcomes reported by other researchers [41]. It is likely that inadequate adherence is a common underlying factor in these cases. Socio-economically vulnerable persons and mobile workers could constitute a reservoir for perpetuation of the HIV epidemic in high-burden communities. With increasing access to ART but inadequate virological suppression in these populations, it is also possible that accumulation of drug-resistant viruses can occur, with risk of onwards transmission [62]. Apart from surveillance, targeted interventions should be considered to improve retention in care, adherence and virological outcomes in individuals with socio-economic risk factors [7, 63]. Alternative and more flexible models for drug dispensing could be effective interventions and are currently under investigation [64, 65].

Our overall study population corresponds well with recent demographic HIV data in urban parts of Ethiopia [12, 13, 19, 66]. In agreement with other reports, we found younger age and worse perceived health to be independently associated with lack of virological suppression [9–11, 67, 68]. Interestingly, we did not find an association between gender and lack of virological suppression in multivariable analysis, as previously reported [10, 11, 67]. Gender-specific analysis showed that several socio-economic and behavioral variables, including high alcohol consumption, were associated with unsuppressed VL in men. Furthermore, significantly higher proportions of men than women had mobile work in multiple locations overall (16.0% vs. 2.1%; p<0.001). These findings suggest that the higher rates of virological failure previously observed among men might at least partially be explained by underlying socio-economic conditions [8, 46].

In the overall study population, incomplete virological suppression was not independently associated with factors indicating sexual risk behavior or substance use. We further assessed mental health status, using the PHQ-2 and PHQ-9 instruments. Although these tools have been validated in Ethiopia and East Africa, we found unexpectedly low rates of symptoms of depressive disorder in our population compared to previous studies among PLHIV in this region [34, 69, 70]. It is possible that the cut-off value ($\geq$3) used in the PHQ-2 was too insensitive for detecting major depressive disorder in this study population.

This study has certain limitations. Since the majority of participants were urban residents, the associations found might not be translated to rural inhabitants. While the numbers of female commercial sex workers and male transport workers in our uptake area are higher than in other parts of Ethiopia, the total number of study participant belonging to key populations was small [22, 23]. This could explain the relatively low number of persons reporting high-risk behavior. Furthermore, selection bias of included participants cannot be excluded, both with regard to participants who could not be reached and for those who declined participation. In addition, the study design did not allow estimation of the proportion of all persons starting ART at the study clinics who remained in care, neither of those who had received routine VL testing according to the Ethiopian guidelines. Since no previous studies with a similar objective had been performed in the area, rates of several investigated variables were unknown and, in several instances, found to be lower than expected. It is therefore possible that the study was underpowered for certain potential associations, in particular with regard to risk behavior.

## Conclusion

In this case-control study of ART recipients in Ethiopia, low relative wealth and geographic work mobility were associated with lack of virological suppression. These findings imply that

increased attention to socio-economic factors is required in order to obtain better and durable outcomes of ART programmes in sub-Saharan Africa.

## Supporting information

**S1 Table. Comparison of participants who could be reached and included, with eligible participants who could not be reached.**
(DOCX)

**S2 Table. Gender-aggregated univariate data of cases and controls.**
(DOCX)

**S3 Table. Gender-aggregated univariate data of cases and controls.**
(DOCX)

**S4 Table. Gender-aggregated univariate data of cases and controls.**
(DOCX)

**S5 Table.**
(DOCX)

## Acknowledgments

The authors would like to thank the study participants, as well as the staff at the ART clinics for their help with conducting this study. We further acknowledge Zelalem Ketema for his important role in data collection as well as Gadisa Labata, Hakenew Tarku, and other members of the Lund University Adama research team. We would also like to thank the Oromia Regional Health Bureau and the Armauer Hansen Research Institute for their assistance and collaboration.

## Author Contributions

**Conceptualization:** Martin Plymoth, Eduard J. Sanders, Elise M. Van Der Elst, Patrik Medstrand, Fregenet Tesfaye, Taye Balcha, Per Björkman.

**Data curation:** Martin Plymoth.

**Formal analysis:** Martin Plymoth, Niclas Winqvist.

**Funding acquisition:** Per Björkman.

**Investigation:** Martin Plymoth.

**Methodology:** Martin Plymoth, Eduard J. Sanders, Elise M. Van Der Elst, Fregenet Tesfaye, Niclas Winqvist, Per Björkman.

**Project administration:** Per Björkman.

**Writing – original draft:** Martin Plymoth, Per Björkman.

**Writing – review & editing:** Martin Plymoth, Eduard J. Sanders, Elise M. Van Der Elst, Patrik Medstrand, Fregenet Tesfaye, Niclas Winqvist, Taye Balcha, Per Björkman.

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
