## [Decision Letter · Decision Letter 0]

19 Aug 2020

PONE-D-20-07483

Socio-economic condition and lack of virological suppression among adults and adolescents receiving antiretroviral therapy in Ethiopia

PLOS ONE

Dear Dr. Plymoth,

Thank you for submitting your manuscript to PLOS ONE. After careful consideration, we feel that it has merit but does not fully meet PLOS ONE’s publication criteria as it currently stands. Therefore, we invite you to submit a revised version of the manuscript that addresses the points raised during the review process.

We look forward to receiving your revised manuscript.

Kind regards,

Jianhong Zhou

Associate Editor

PLOS ONE

Journal Requirements:

Additional Editor Comments (if provided):

Reviewers' comments:

Reviewer's Responses to Questions

**Comments to the Author**

1. Is the manuscript technically sound, and do the data support the conclusions?

Reviewer #1: Yes

Reviewer #2: Yes

2. Has the statistical analysis been performed appropriately and rigorously? 

Reviewer #1: Yes

Reviewer #2: Yes

3. Have the authors made all data underlying the findings in their manuscript fully available?

Reviewer #1: Yes

Reviewer #2: Yes

4. Is the manuscript presented in an intelligible fashion and written in standard English?

Reviewer #1: Yes

Reviewer #2: Yes

5. Review Comments to the Author

Reviewer #1: I read with interest this manuscript.

I think that is well wrote and with a good idea research from an important setting (Ethiopia)

Only some suggestions:

1. Introduction: line 44 is the definition of social determinants of health (cite Marmot et al on this issue)

join line 55-56; delete line 71-72

2. Methods: write the time of the study.

3. Results:well wrote. Delete line189

4. Discussion: well wrote. Compare with other study of low setting (ec cite Prevalence and Predictors of Malaria in Human Immunodeficiency Virus Infected Patients in Beira, Mozambique. Int J Environ Res Public Health. 2018;15(9):2032. Published 2018 Sep 17 and The At Risk Child Clinic (ARCC): 3 Years of Health Activities in Support of the Most Vulnerable Children in Beira, Mozambique. Int J Environ Res Public Health. 2018;15(7):1350. Published 2018 Jun 27. doi:10.3390/ijerph15071350) who underline how women and child are the most vulenrable group and the comorbidity and co-infection with Malaria can be devasting in people with HIV.

Furhermore, as for HIV also TB responde to social determinats of health. (doi:10.1016/j.tube.2017.01.002 and doi:10.1186/s13104-018-3209-9)

5. Conclusion: improve with a advocy that only pharmacological approuch is not sufficient to control burden of HIV

6. Table and statistical analysis are very well done. Congratulations

Reviewer #2: Introduction

Since such factors might be associated

66 with risk of HIV transmission, we also investigated whether lack of virological

67 suppression during ART is associated with high-risk sexual behaviour.

Reviewer comments

Yes, socioeconomic factors might be associated with HIV transmission, but I don’t see why you use that fact to support your secondary aim, you cannot use that fact to motivate your reason for investigating the association between virological suppression during ART and high-risk behaviour (you may look for a reference that supports this association), rather simply state that: A secondary aim of our study was to investigate the association between virological suppression and high-risk sexual behaviour.

For this purpose,

68 we performed a case-control study of ART recipients at Ethiopian public health

69 facilities, comparing persons with VL>1000 copies/ml to those with virological

70 suppression.

Reviewer comments

To me, that statement makes it as if you are conducting a solely comparative study, it completely drifts away from your title and the introduction. Nowhere in the introduction do you talk about virologically suppressed Vs non-suppressed. Besides the fact that you are focusing attention on the association between lack of virological suppression and socioeconomic condition, the comparative factor strengthens your study even further. If you are comparing, this aspect of your study must “come out” in the study title, introduction and the aims. WHY ARE YOU COMPARING THE VIROLOGICALLY SUPPRESSED VS. NON SUPPRESSED? To have a referral point? Then say so. WOULD YOU HAVE FOUND DIFFERENT RESULTS IF YOU HAD NOT COMPARED, i.e., if you had only sought for associations between socioeconomic conditions and those with VL>1000 copies/ml?

Alternatively, your sentence could read: In order to understand the socioeconomic conditions impacting on virological suppression, we performed a case-control study of ART recipients at Ethiopian public health facilities, comparing persons with VL>1000 copies/ml to those with virological suppression.

Methods, results and ethical considerations

Reviewer comments

Comprehensive and well-articulated.

Discussion

Reviewer comments

The discussion is adequate. However, you fail to compare your findings with other studies that have also ‘compared’ virologically suppressed vs. non-suppressed.

General comments

All in all, this is a good study, and I praise you for the good work.

6. PLOS authors have the option to publish the peer review history of their article (what does this mean?). If published, this will include your full peer review and any attached files.

Reviewer #1: **Yes: **Francesco Di Gennaro

Reviewer #2: **Yes: **Smart Z Mabweazara

---

## [Author Response · Author response to Decision Letter 0]

6 Oct 2020

Dear Dr. Jianhong Zhou, Dr. Francesco Di Gennaro and Dr. Smart Z Mabweazara,

We appreciate your review of our manuscript and the comments provided. We have addressed these comments point by point. For most comments, we have followed the advice given by the reviewers. However, for certain comments we prefer to keep the original formulations in the submitted manuscript, and we have explained these reasons in the replies inserted in the attached rebuttal letter.

We hope that our revision of the manuscript will be to your satisfaction, as well as the replies provided, and that the manuscript will meet criteria for publication in PLOS One. 

Yours sincerely,

Martin Plymoth and Per Björkman

---

## [Decision Letter · Decision Letter 1]

18 Nov 2020

PONE-D-20-07483R1

Socio-economic condition and lack of virological suppression among adults and adolescents receiving antiretroviral therapy in Ethiopia

PLOS ONE

Dear Dr. Plymoth,

Thank you for submitting your manuscript to PLOS ONE. After careful consideration, we feel that it has merit but does not fully meet PLOS ONE’s publication criteria as it currently stands. Therefore, we invite you to submit a revised version of the manuscript that addresses the points raised during the review process.

We look forward to receiving your revised manuscript.

Kind regards,

Kennedy Otwombe

Academic Editor

PLOS ONE

Reviewers' comments:

Reviewer's Responses to Questions

**Comments to the Author**

1. If the authors have adequately addressed your comments raised in a previous round of review and you feel that this manuscript is now acceptable for publication, you may indicate that here to bypass the “Comments to the Author” section, enter your conflict of interest statement in the “Confidential to Editor” section, and submit your "Accept" recommendation.

Reviewer #1: All comments have been addressed

Reviewer #3: All comments have been addressed

2. Is the manuscript technically sound, and do the data support the conclusions?

Reviewer #1: Yes

Reviewer #3: Yes

3. Has the statistical analysis been performed appropriately and rigorously? 

Reviewer #1: Yes

Reviewer #3: I Don't Know

4. Have the authors made all data underlying the findings in their manuscript fully available?

Reviewer #1: Yes

Reviewer #3: Yes

5. Is the manuscript presented in an intelligible fashion and written in standard English?

Reviewer #1: Yes

Reviewer #3: Yes

6. Review Comments to the Author

Reviewer #1: Authors improve their manuscript that now can be publish

I appreciate the research question and also the setting of study

Reviewer #3: Pages 15-18 mentions several significant p-values, yet no mention applying Holm-Sidak. Exactly how were these multiple p-values addressed to minimize Type 1 error?

Table 4 p value are those post-Holm Sidak? Better to see the original p-values apply the Holm-Sidak, then show the outcome.

Other tables show the p-values as well, we they not included in the Holm-Sidak correction?

The Holm-Sidak application confuses. Please clarify…

There are still problems with the abbreviations, where a number of them were given at first mention without presenting the meaning.

7. PLOS authors have the option to publish the peer review history of their article (what does this mean?). If published, this will include your full peer review and any attached files.

Reviewer #1: No

Reviewer #3: No

---

## [Author Response · Author response to Decision Letter 1]

23 Nov 2020

Please se attached document "rebuttal letter/response to reviewers"

---

## [Editor Report · Decision Letter 2]

3 Dec 2020

Socio-economic condition and lack of virological suppression among adults and adolescents receiving antiretroviral therapy in Ethiopia

PONE-D-20-07483R2

Dear Dr. Plymoth,

We’re pleased to inform you that your manuscript has been judged scientifically suitable for publication and will be formally accepted for publication once it meets all outstanding technical requirements.

Kind regards,

Kennedy Otwombe

Academic Editor

PLOS ONE

Additional Editor Comments (optional):

NA
---

## [Editor Report · Acceptance letter]

7 Dec 2020

PONE-D-20-07483R2 

Socio-economic condition and lack of virological suppression among adults and adolescents receiving antiretroviral therapy in Ethiopia 

Dear Dr. Plymoth:

I'm pleased to inform you that your manuscript has been deemed suitable for publication in PLOS ONE. Congratulations! Your manuscript is now with our production department. 

Kind regards, 

on behalf of

Dr. Kennedy Otwombe 

Academic Editor

PLOS ONE